# Risk of adverse pathological features for intermediate risk prostate cancer: Clinical implications for definitive radiation therapy

**Hong Zhang**[1]\*, **Christopher Doucette**[1], **Hongmei Yang**[2], **Sanjukta Bandyopadhyay**[2], **Craig E. Grossman**[3], **Edward M. Messing**[4], **Yuhchyau Chen**[1]

1 Department of Radiation Oncology, University of Rochester Medical Center, Rochester, NY, United States of America, 2 Department of Biostatistics and Computational Biology, University of Rochester Medical Center, Rochester, NY, United States of America, 3 Department of Radiation Oncology, Stony Brook University Hospital, Stony Brook, NY, United States of America, 4 Department of Urology, University of Rochester Medical Center, Rochester, NY, United States of America

\* hong_zhang@urmc.rochester.edu

## Abstract

### Background

Intermediate risk prostate cancer represents a largely heterogeneous group with diverse disease extent. We sought to establish rates of adverse pathological features important for radiation planning by analyzing surgical specimens from men with intermediate risk prostate cancer who underwent immediate radical prostatectomy, and to define clinical pathologic features that may predict adverse outcomes.

### Materials and methods

A total of 1552 men diagnosed with intermediate risk prostate cancer who underwent immediate radical prostatectomy between 1/1/2005 and 12/31/2015 were reviewed. Inclusion criteria included available preoperative PSA level, pathology reports of transrectal ultrasound-guided prostate biopsy, and radical prostatectomy. Incidences of various pathological adverse features were evaluated. Patient characteristics and clinical disease features were analyzed for their predictive values.

### Results

Fifty percent of men with high risk features (defined as PSA >10 but <20 or biopsy primary Gleason pattern of 4) had pathological upstage to T3 or higher disease. The incidence of upgrade to Gleason score of 8 or higher and the incidence of lymph node positive disease was low. Biopsy primary Gleason pattern of 4, and PSA greater than 10 but less than 20, affected adverse pathology in addition to age and percent positive biopsy cores. Older age and increased percentage of positive cores were significant risk factors of adverse pathology.

**Data Availability Statement:** All relevant data are within the manuscript and its Supporting Information files.

**Funding:** The author(s) received no specific funding for this work.

**Competing interests:** The authors have declared that no competing interests exist.

## Conclusion

Our findings underscore the importance of comprehensive staging beyond PSA level, prostate biopsy, and CT/bone scan for men with intermediate risk prostate cancer proceeding with radiation in the era of highly conformal treatment.

## Introduction

According to guidelines published by the National Comprehensive Cancer Network (NCCN), intermediate risk prostate cancer (PCa) is defined as having at least one of the following features: cT2b to cT2c, Gleason score (GS) 7 (3+4/4+3), and PSA 10 to 20 ng/ml [1]. This is a large heterogeneous group that has diverse outcomes. Many attempts have been made to further stratify this risk grouping in order to tailor appropriate treatments. For example, NCCN guidelines currently divide the intermediate risk group into "favorable" and "unfavorable" subgroups [1]. Favorable intermediate risk PCa is defined as having only one of the following features: cT2b-cT2c, GS 3+4 = 7, PSA 10 to 20 ng/ml, and percentage of positive biopsy cores (PPC) <50%. Unfavorable intermediate risk disease is defined as having GS 4+3 = 7 disease, or having at least two of the following risk features: cT2b-cT2c, GS 3+4 = 7, PSA 10 to 20 ng/ml, and PPC > = 50%.

Management of intermediate risk PCa ranges from active surveillance to prostatectomy, external beam radiation (EBRT) with or without androgen deprivation (ADT), combined external beam radiation with brachytherapy boost with or without ADT, or brachytherapy alone [1]. There is also emerging evidence to support the usage of stereotactic body radiotherapy (SBRT) for this disease group [2–7]. It remains a major challenge to choose the most appropriate treatment for men with intermediate risk PCa. Various reports in the literature have shown that the current method of risk characterization based on PSA level, prostate biopsy, and imaging modality (including CT and bone scan) is not adequate in predicting true disease extent such as extracapsular extension, seminal vesicle and/or bladder neck invasion, or disease fixed/extending into surrounding normal structures [8–18].

For men receiving curative-intent radiotherapy (RT), treatment success relies on accurate delineation of target volumes. Recent technology improvement in treatment planning and delivery with image guidance and intensity-modulated RT has improved treatment precision by reducing the treatment margins, therefore limiting radiation dose to surrounding normal tissues. However, the reduction of margins without considering the pattern and extent of the microscopic spread of disease may negatively influence outcomes [19–22].

The purpose of this study was to estimate the frequency of locoregional disease extension through analyses of pathologic reports in men with intermediate risk PCa undergoing immediate prostatectomy. We anticipated that valuable information could be derived from analyzing these surgical reports, which could result in more precise radiation treatment plans. Patient demographics and tumor characteristics considered important in predicting the presence of clinically significant disease were also included in our analyses.

## Materials and methods

### Ethics statement

This retrospective review study was approved by our institutional research subjects review board (IRB) at the University of Rochester (Approve Number: RSRB00057633). The IRB waived

the requirement for informed consent. The information collected about the study subjects already existed in the medical record. Protected health information was not used by or disclosed to any other person except authorized investigators and was fully anonymized before analysis.

## Study population

Between 1/1/2005 and 12/31/2015, the clinical information and pathology report of men diagnosed with intermediate risk PCa and who underwent radical prostatectomy at our institution were reviewed. Among them, 1552 men were included in the analyses (S1 Dataset). Inclusion criteria were clinical intermediate risk PCa (based on PSA, biopsy pathology, physical exam, staging bone scan and CT pelvis) and available pathology reports of prostatectomy. A total of 1376 men had nodal sampling/dissection at the time of surgery. Clinical T subclassification (T2a-2c) was not consistently recorded, so clinical T stage was not used for analysis. As a result, we were not able to divide our study cohort into favorable and unfavorable groupings according to NCCN guidelines.

## Review of surgical pathology

Adverse pathological features were defined as the presence of extracapsular extension, seminal vesicle invasion, bladder neck invasion, disease fixed/extending into other surrounding normal structures, GS 8 or higher, and pathological nodal positivity.

## Statistical analysis

Statistical comparisons between groups were performed using Chi-square test or Student t test as appropriate. The corresponding non-parametric version of Fisher Exact test or Wilcoxon Rank Sum test were used for confirmation when sample sizes were small. Logistic regression was used to study the effects of demographics and clinical variables on the outcomes, including upstage to pathological T3 or higher, pathological node-positive (PNP) disease, and total surgical GS upgrade to $> = 8$. Odds ratios with 95% confidence intervals were reported to represent the likelihood of each characteristic affecting outcomes relative to its reference level. The clinical outcome of time to PSA failure was defined as the time to first of two consecutive PSA levels to $> = 0.2$ ng/ml after radical prostatectomy. The marginal effects of several variables, including biopsy GS, pathological upstage, percentage of positive biopsy cores (PPC), and upgrade on time to PSA failure were presented by Kaplan-Meier infection-free probability curves and tested by Log-rank test. A Cox proportional hazards regression model was further adapted to examine the effect of each covariate, while adjusting for the effect of other covariates. The proportional hazards assumption and possible outliers were visually examined through Schoenfeld and deviance residual plots. Hazard ratios with 95% confidence intervals were presented for the assessment of risk. All the multivariate models were coupled with stepwise variable selection for improved accuracy. Statistical significance was defined at a level of P $< = 0.05$.

All statistical analyses were conducted using Version 9.4 of the SAS System for Windows (SAS Institute Inc., Cary, NC, USA).

# Results

## Study characteristics

Baseline patient demographics and PCa characteristics are shown in Table 1. The majority of men (88.9%) in this cohort were ≤70 years of age (median age of 62); 87% had a PSA ≤ 10; 26.5% had GS 4+3 = 7 disease; 99.6% had cT2 disease. Only 3.2% of patients received ADT.

**Table 1. Baseline patient characteristics.**

| | Number of patients | Percent of patients |
|---|---|---|
| *Year of diagnosis* | | |
| 2005--2010 | 709 | 45.7 |
| 2011--2015 | 843 | 54.3 |
| *Age* | | |
| Median | 62 | |
| < = 60 | 639 | 41.2 |
| 61-70 | 741 | 47.7 |
| >70 | 172 | 11.1 |
| *Race* | | |
| White | 1347 | 86.8 |
| African | 119 | 7.7 |
| Others (including unknown) | 85 | 5.4 |
| *PSA* | | |
| < = 10 | 1350 | 87.0 |
| >10 but < = 20 | 202 | 13.0 |
| *Clinical Stage* | | |
| T1 | 6 | 0.4 |
| T2 | 1546 | 99.6 |
| *Gleason score* | | |
| 3+3 | 276 | 17.8 |
| 3+4 | 865 | 55.7 |
| 4+3 | 411 | 26.5 |

## The incidence of pathological tumor upstage and upgrade

Overall, 37.3% cases had pathological T3 or higher disease (PTUS) among the 1552 men in the study (specifically 36.9% with extracapsular extension and 6.9% with seminal vesicle invasion). The frequency of PTUS was higher in the higher PSA group (50% for PSA >10 but <20 vs. 35% for PSA < = 10, p<0.001), higher GS groups (50% for GS 4+3 = 7 vs. 36% for GS 3+4 = 7 vs. 24% for GS 3+3 = 6, p<0.001) (Fig 1A).

The incidence of GS upgrade (PGUG) was noted to be 24% for the entire cohort. The incidence of PGUG to 8 or higher was 4%; it was higher in the higher PSA group (7% for PSA >10 but <20 vs. 3% for PSA < = 10, p = 0.02), primary Gleason pattern of 4 at biopsy (9% vs. 2% for primary Gleason pattern 3 at biopsy, p<0.001) (Fig 1B).

Only 4% of patients were noted to have PNP disease in the cohort. The incidence of PNP was higher in the higher PSA group (7% for PSA >10 but <20 vs. 4% for PSA < = 10, p<0.05), primary Gleason pattern of 4 at biopsy (7% for GS 4+3 = 7 vs. 3% for GS 3+4 = 7 vs. 1% for GS 3+3 = 6 at biopsy, p<0.001) (Fig 1C).

## Risk factors for upstage and upgrade

The univariate analysis noted that race was not a significant risk factor for a PTUS, or PGUG to GS > = 8 in this cohort (p = 0.32 and 0.12, respectively).

Univariate analyses showed PSA level and biopsy GS were significant predictors for PTUS (p<0.001 and p<0.001 respectively) or PGUG to 8 or higher disease (p = 0.02, p<0.01, and p<0.001 respectively). PSA level and biopsy GS were significant predictors for nodal positive disease (p = 0.038 and p<0.001, respectively).

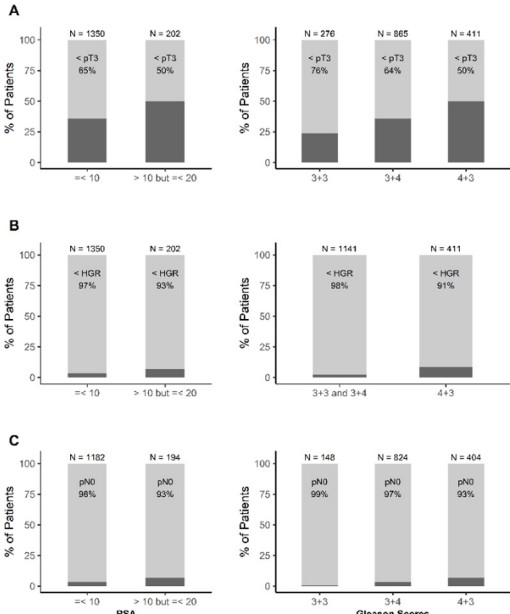

**Fig 1. Incidence of adverse pathological features among subsets of patients.** A) percentage of cases with pathological upstage to pT3 or higher disease among men with different PSA levels and GS at the biopsy. B) the percentage of cases with pathological upgrade to GS 8 or higher among men with different PSA levels and GS at the biopsy. C) the percentage of cases with pathological node positive disease among men with different PSA levels and GS at the biopsy.

Multivariate analyses (Fig 2A) showed that age and PPC were statistically significant factors for PTUS. The risk of PTUS increased 37% with every 10-year increase of age (p<0.001), and 31% with 10% increase in PPC (p<0.001). PSA level and primary Gleason pattern at biopsy interactively affected PTUS. Specifically, when PSA < = 10, primary Gleason pattern of 4 at biopsy, compared with the primary Gleason pattern of 3, increased the risk of PTUS by 32% (p = 0.003). When primary Gleason pattern of 3 at biopsy, PSA level >10 but <20 resulted in a 19% higher risk of PTUS comparing with PSA < = 10 but with marginal significance (p = 0.056).

PSA level and primary Gleason pattern at biopsy interactively affected PGUG to 8 or higher. Specifically, when biopsy primary Gleason pattern was 3, a 41% increased risk of PGUG to 8 or higher was observed when PSA >10 but <20, compared with PSA < = 10 (p = 0.04) (Fig 2B). Biopsy primary Gleason pattern of 4, comparing with Gleason pattern of 3, significantly

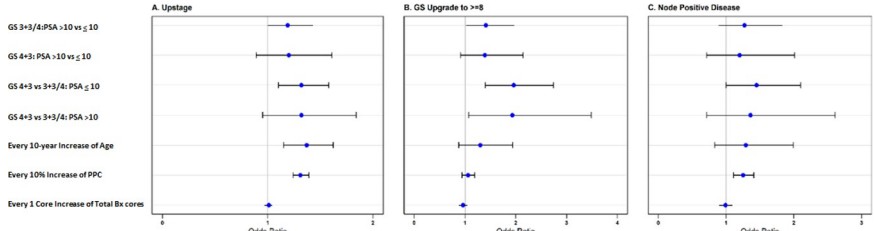

**Fig 2. Multivariate analyses of factors predictive of pathological upstage, upgrade and node positive disease.** Odds ratios (with 95% confidence intervals in S1 Table) of factors predictive of A) pathological upstage to pT3 or higher disease, B) pathological upgrade to GS 8 or higher, and C) pathological node positive disease. GS- Gleason Scores. Bx = biopsy. PPC = percent positive biopsy cores.

increased the risk of PGUS to 8 or higher regardless of PSA level (96% and 93% increased risk of PGUG to 8 or higher when PSA >10 but <20 and PSA < = 10 respectively, p<0.001 and p = 0.028 respectively). Age or PPC was not significantly associated with PTUG to 8 or higher (Fig 2B).

There was 25% increased in the risk of PNP disease with every 10% increase of PPC (p<0.001) (Fig 2C). Primary biopsy GS and PSA level interactively influenced the risk of PNP disease. Primary Gleason pattern affected PNP disease differently only when PSA < = 10. Risk of PNP disease increased by 45% when biopsy primary Gleason pattern of 4, comparing with primary Gleason pattern of 3 at biopsy, when PSA< = 10 (p = 0.047). Age was not significantly associated with PNP disease.

## Clinical outcomes

Cox proportion model was used to study the joint effects of the following factors on long-term PSA failure: age, primary Gleason pattern at biopsy, PSA level, the interaction between PSA and Primary Gleason pattern at biopsy, PTUS, PGUG, GS at prostatectomy, PTG pattern, upgrade to higher pathological GS than at biopsy, PPC, total biopsy cores, extracapsular extension, positive margin, seminal vesicle invasion, lymphovascular invasion, perineural invasion, or PNP. Model fitting results suggested primary Gleason pattern of 4 at biopsy, when compared with primary Gleason pattern of 3, had a 2.35 fold increased risk of shorter time to PSA failure (p<0.001) (Fig 3). PTUS and PGUG resulted in 1.52 and 1.33 fold increased risk of shorter time to PSA failure (p = 0.005 and p = 0.001 respectively). Furthermore, there was a 10% increased risk of early PSA failure with each increased core of total biopsy (p = 0.001).

## Prediction of seminal vesicle invasion using Partin tables

Among 1552 men with intermediate risk PCa, 118 was noted to have seminal vesicle disease involvement at the time of prostatectomy. We calculated the risk of seminal vesicle using Partin tables based on available clinical information in 108 out of 118 men [23]. Thirty-four men (35.1%) had < = 5% estimated risk of seminal vesicle disease involvement in this cohort of pathologically confirmed disease (Fig 4).

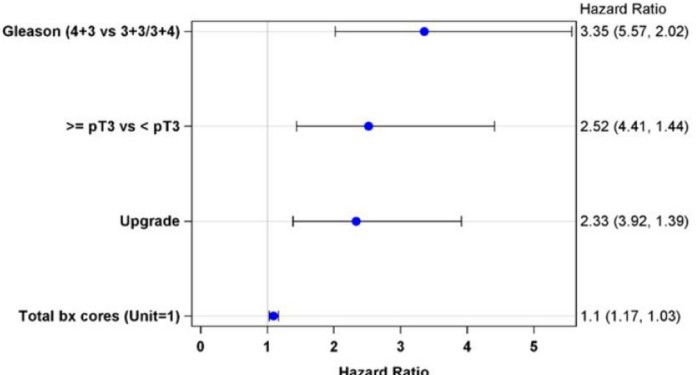

**Fig 3. Multivariate analyses of factors predictive of time to PSA failure.** Odds ratios (with 95% confidence interval in parenthesis) of factors predictive of time to PSA failure.

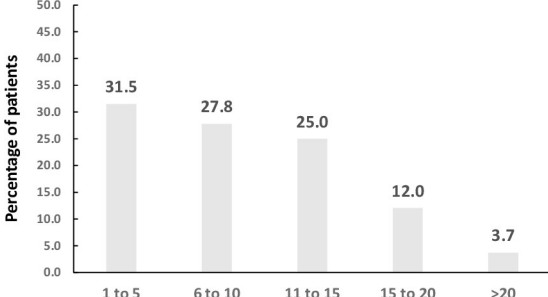

**Fig 4. Estimated risk of seminal vesicle involvement using Partin tables in 108 men with confirmed seminal vesicle disease at the time of prostatectomy.**

## Discussion

In our cohort of men with intermediate risk PCa, there was a subgroup of patients that had more advanced disease than what clinical stage indicated. Among 1552 men with intermediate risk PCa who underwent immediate prostatectomy at our institution, 50% with PSA >10 or primary Gleason pattern of 4 at biopsy had PTUS. Older age was an adverse risk factor of PTUS. When controlling other factors, PSA level and primary Gleason pattern interactively influenced PTUS.

Numerous studies have reported upstage incidence among patients with localized PCa. Most studies evaluated men with either low or favorable intermediate risk disease [6–17]. The incidence rate of pathological upstage to pT3 or higher disease among men with intermediate risk disease in our study was about 37%; the incidence rate of pathological upgrade to Gleason GS 8 or higher was 4%. A recent study from Johns Hopkins University reported near 25% upgrade and upstage among low volume intermediate risk PCa (defined as 1–2 positive cores, Gleason 3 + 4 = 7, PSA <20) at their institution [17]. Martin et al. have shown that 14% of unfavorable intermediate risk prostate cancer upgraded to GS 8 or higher disease [18]. The difference between our cohort and the above studies was likely due to the inclusion of favorable and unfavorable intermediate risk disease in our cohort.

We report 50% incidence of pT3 or higher disease among men with either PSA >10 but <20 or biopsy primary Gleason pattern of 4 intermediate risk PCa. In fact, PSA level and biopsy primary Gleason pattern together were significant risk factors of PTUS. These findings are important when radiation is the primary therapy for this risk group of men with intermediate risk PCa. Appropriate clinical target volume (CTV) should cover the area at risk of microscopic disease spread for these patients. Pathological evaluations provided guidance for CTV definition when there was an extracapsular extension or seminal vesicle invasion [19–22]. A retrospective review by Chao et al. found that more than 20% of patients with PSA >10 or biopsy GS > = 7 had an extracapsular extension extending beyond 4 to 5 mm [19]. When there was disease involvement in the seminal vesicle, 10% of cases had disease extending beyond the proximal 2 cm [22]. These reports, together with our findings of high PTUS incidence among a selected group, provided a cautionary note when reducing CTV for intermediate risk PCa with high risk features (PSA >10, biopsy primary Gleason pattern of 4 and the high percentage of positive cores).

It is important to point out that none of our patients underwent pre-treatment multiparementric (mp) MRI of prostate as part of the staging workup. mp MRI should be considered as

part of staging workup before proceeding with radiation. The efficacy of mp MRI in defining grade and prostate cancer disease extent has been elucidated by various reports [24–27]. Pooled sensitivity and specificity of mpMRI to detect prostate cancer with GS > = 5 disease were 0.89 (95% CI 0.86 to 0.92) and 0.73 (95% CI 0.60 to 0.83) using Performance of Prostate Imaging Reporting and Data System Version 2 (PI-RADS v2) [28]. Reported sensitivity and specificity of mpMRI to detect extracapsular extension were 60 to 81% and 75 to 78%, respectively [29, 30]; reported sensitivity and specificity of mpMRI to detect seminal vesicle disease were 73% and 95%, respectively [31]. After perspectively reviewing 377 cases of intermediate risk prostate cancer treated with radical prostatectomy, Roumiguie et al. have shown that integrating mpMRI and targeted biopsy, to standard intermediate risk group (IRC, including PSA, clinical stage, Gleason Grade, and percentage of positive cores on biopsy), outperformed standard one by 15% in predicting adverse pathological features including pT3-4 and Gleason Grade group 3 or more disease [32]. Specifically, IRC with mpMRI and targeted biopsy identified 71.7% of pT3-4 and/or N1 disease compared with 62.2% by standard IRC alone. Further, IRC with mpMRI and targeted biopsy predicted 62.3% of grade group 3 or higher disease compared with 43.3% by standard IRC.

In our study, the incidence of PNP disease was low at 4% in the cohort with lymph node sampling. Even among the group with primary Gleason pattern of 4 disease, incidence was only 7%. However, the actual incidence might be higher than this due to sampling errors secondary to the extent of nodal sampling/dissection at the time of surgery.

Our study has several weaknesses due to the retrospective design and the inherent deficiency of data reported in medical records. Clinical T stage, specifically T2 subgrouping, was not clearly defined in the medical records and we had no way to independently verify these. There was a wide range of the total number of prostate biopsy cores reported. Further, an international consensus update of Gleason grading in 2014 resulted in some changes of Gleason pattern 3 and 4 classifications. Although this grading update was unlikely to significantly change the findings in our study, the precise impact was impossible to define.

Despite the above limitations, our large retrospective review has shown a near 50% incidence of adverse pathology in men with intermediate risk PCa with high risk features. PSA>10, biopsy primary Gleason pattern of 4, age, and increased tumor burden as suggested by the percentage of positive cores were significant predictors of pathological upstage. Our findings underscore the importance of improved clinical staging and careful design of target volume for radiation in the era of highly conformal targeted radiation, especially SBRT, for a select group of patients with intermediate risk PCa.

## Supporting information

**S1 Table. 95% confidence interval of Fig 2.**
(DOCX)

**S1 Dataset.**
(XLSX)

## Acknowledgments

The authors thank Mrs. Laura Finger for editorial assistance.

## Author Contributions

**Conceptualization:** Hong Zhang, Craig E. Grossman, Edward M. Messing.

**Data curation:** Hong Zhang, Christopher Doucette, Hongmei Yang, Sanjukta Bandyopadhyay, Craig E. Grossman.

**Formal analysis:** Hong Zhang, Christopher Doucette, Hongmei Yang, Sanjukta Bandyopadhyay, Edward M. Messing, Yuhchyau Chen.

**Investigation:** Hong Zhang, Christopher Doucette, Craig E. Grossman, Edward M. Messing, Yuhchyau Chen.

**Methodology:** Hong Zhang, Christopher Doucette, Hongmei Yang, Sanjukta Bandyopadhyay, Craig E. Grossman, Edward M. Messing, Yuhchyau Chen.

**Validation:** Hong Zhang.

**Writing – original draft:** Hong Zhang.

**Writing – review & editing:** Hong Zhang, Christopher Doucette, Hongmei Yang, Sanjukta Bandyopadhyay, Craig E. Grossman, Edward M. Messing, Yuhchyau Chen.

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
