## [Decision Letter · Decision Letter 0]

11 Mar 2021

PONE-D-21-03241

Risk of adverse pathological features for intermediate risk prostate cancer: clinical implications for definitive radiation therapy

PLOS ONE

Dear Dr. Zhang,

Thank you for submitting your manuscript to PLOS ONE. After careful consideration, we feel that it has merit but does not fully meet PLOS ONE’s publication criteria as it currently stands. Therefore, we invite you to submit a revised version of the manuscript that addresses the points raised during the review process.

We look forward to receiving your revised manuscript.

Kind regards,

Qinghui Zhang

Academic Editor

PLOS ONE

Journal Requirements:

2. Please note that PLOS does not permit references to “data not shown.” Authors should provide the relevant data within the manuscript, the Supporting Information files, or in a public repository. If the data are not a core part of the research study being presented, we ask that authors remove any references to these data.

Reviewers' comments:

Reviewer's Responses to Questions

**Comments to the Author**

1. Is the manuscript technically sound, and do the data support the conclusions?

Reviewer #1: Yes

Reviewer #2: Yes

2. Has the statistical analysis been performed appropriately and rigorously? 

Reviewer #1: Yes

Reviewer #2: Yes

3. Have the authors made all data underlying the findings in their manuscript fully available?

Reviewer #1: No

Reviewer #2: Yes

4. Is the manuscript presented in an intelligible fashion and written in standard English?

Reviewer #1: Yes

Reviewer #2: Yes

5. Review Comments to the Author

Reviewer #1: This is a well-written retrospective review of patients who underwent definitive surgery for intermediate risk prostate cancer. The study highlights the importance of adequate pre-clinical staging and demonstrates further benefit for the use of MP-MRI prior to definitive-intent treatment. The statistical analysis have been performed appropriately, and the author's conclusions are supported by them.

Minor stylistic/grammar comments:

1) Consider rewording "Per NCCN guidelines, intermediate risk prostate cancer (PCa) has been defined as having at least one of the following features: cT2b to cT2c, Gleason score (GS) 7 (3+4/4+3), and PSA 10 to 20 ng/ml." as "According to guidelines published by the National Comprehensive Cancer Network (NCCN), intermediate risk prostate cancer (PCa) is defined as having at least one of the following features: cT2b to cT2c, Gleason score (GS) 7 (3+4/4+3), and PSA 10 to 20 ng/ml."

2) Consider rewording "For men receiving definitive curative intent radiotherapy (RT), the success of treatment relies on accurately defining the target volume in order to ensure disease control and reduce toxicity. Recent technology improvement in treatment planning and delivery with image guidance and intensity-modulated RT has improved treatment precision by reducing the treatment margins, therefore limiting radiation dose to surrounding normal tissues. However, the reduction of margins without considering the pattern and extent of microscopic spread of disease may negatively influence the outcome." to "For men receiving curative-intent radiotherapy (RT), treatment success relies on accurate delineation of target volumes. Recent technology improvement in treatment planning and delivery with image guidance and intensity-modulated RT has improved treatment precision by reducing the treatment margins, therefore limiting radiation dose to surrounding normal tissues. However, the reduction of margins without considering the pattern and extent of microscopic spread of disease may negatively influence outcomes."

Minor comments:

1) Please describe pre-treatment workup in greater detail in the Methods section. In the discussion, it is stated that *none* of the patients underwent MP-MRI prior to treatment. Did pre-treatment work-up solely consist of rectal exam, PSA, and TRUS biopsy/findings? This is crucial to the results of the study.

2) In the discussion section, I recommend further discussion on the utility of MP-MRI, including rates of upstaging from these studies.

Overall, the study is well done, and merits publication in a journal.

Reviewer #2: A retrospective review addressing a clinically significant question. A Well written, easy read, and statistically sound manuscript. The format of data presentation in figures is right but needs editing/ improvement. The first two figures look busy and difficult to follow. The authors acknowledge the study's limitations and gaps in the data collection, not an uncommon problem in retrospective studies. This will be an excellent addition to the existing literature in the context of uncertainties of clinical staging and delineation of CTV in intermediate-risk prorate cancer patients.

6. PLOS authors have the option to publish the peer review history of their article (what does this mean?). If published, this will include your full peer review and any attached files.

Reviewer #1: No

Reviewer #2: **Yes: **Naseer Ahmed, MD, FRCPC

Radiation Oncologist, Associate Professor, Max Rady College of Medicine, Faculty of Health Sciences, Department of Radiology, Section of Radiation Oncology, University of Manitoba

Affiliate Scientist, Research Institute in Oncology and Hematology, University of Manitoba

---

## [Author Response · Author response to Decision Letter 0]

7 May 2021

We are resubmitting the manuscript titled “Risk of adverse pathological features for intermediate risk prostate cancer: clinical implications for definitive radiation therapy”. Based on the reviewers’ constructive input, we have revised our report. We have included the reviewers’ comments below, with our accompanying point-by-point responses in bold font.

Reviewer #1: 

1) Consider rewording "Per NCCN guidelines, intermediate risk prostate cancer (PCa) has been defined as having at least one of the following features: cT2b to cT2c, Gleason score (GS) 7 (3+4/4+3), and PSA 10 to 20 ng/ml." as "According to guidelines published by the National Comprehensive Cancer Network (NCCN), intermediate risk prostate cancer (PCa) is defined as having at least one of the following features: cT2b to cT2c, Gleason score (GS) 7 (3+4/4+3), and PSA 10 to 20 ng/ml."

We have revised the manuscript as suggested. 

2) Consider rewording "For men receiving definitive curative intent radiotherapy (RT), the success of treatment relies on accurately defining the target volume in order to ensure disease control and reduce toxicity. Recent technology improvement in treatment planning and delivery with image guidance and intensity-modulated RT has improved treatment precision by reducing the treatment margins, therefore limiting radiation dose to surrounding normal tissues. However, the reduction of margins without considering the pattern and extent of microscopic spread of disease may negatively influence the outcome." to "For men receiving curative-intent radiotherapy (RT), treatment success relies on accurate delineation of target volumes. Recent technology improvement in treatment planning and delivery with image guidance and intensity-modulated RT has improved treatment precision by reducing the treatment margins, therefore limiting radiation dose to surrounding normal tissues. However, the reduction of margins without considering the pattern and extent of microscopic spread of disease may negatively influence outcomes."

We have revised the manuscript as suggested. 

Minor comments:

1) Please describe pre-treatment workup in greater detail in the Methods section. In the discussion, it is stated that *none* of the patients underwent MP-MRI prior to treatment. Did pre-treatment work-up solely consist of rectal exam, PSA, and TRUS biopsy/findings? This is crucial to the results of the study.

Pre-treatment work-up included bone scan and CT of pelvis. Reviewer’s point is well taken and we have added the information in the Methods section. 

2) In the discussion section, I recommend further discussion on the utility of MP-MRI, including rates of upstaging from these studies.

We appreciate reviewer’s suggestion. We have reviewed the literature and discussed the utility of MP-MRI in prostate cancer staging in the discussion section, including the rates of upgrade/upstage based on one of the most recent study. 

Reviewer #2: A retrospective review addressing a clinically significant question. A Well written, easy read, and statistically sound manuscript. The format of data presentation in figures is right but needs editing/ improvement. The first two figures look busy and difficult to follow. The authors acknowledge the study's limitations and gaps in the data collection, not an uncommon problem in retrospective studies. This will be an excellent addition to the existing literature in the context of uncertainties of clinical staging and delineation of CTV in intermediate-risk prorate cancer patients.

We appreciate reviewer’s positive comment. We have made changes in Figure 1 and 2 and added a table as a supplement. We hope that they are now clear.

---

## [Decision Letter · Decision Letter 1]

16 Jun 2021

Risk of Adverse Pathological Features for Intermediate Risk Prostate Cancer: Clinical Implications for Definitive Radiation Therapy

PONE-D-21-03241R1

Dear Dr. Zhang,

We’re pleased to inform you that your manuscript has been judged scientifically suitable for publication and will be formally accepted for publication once it meets all outstanding technical requirements.

Kind regards,

Qinghui Zhang

Academic Editor

PLOS ONE

Additional Editor Comments (optional):

Reviewers' comments:

Reviewer's Responses to Questions

**Comments to the Author**

1. If the authors have adequately addressed your comments raised in a previous round of review and you feel that this manuscript is now acceptable for publication, you may indicate that here to bypass the “Comments to the Author” section, enter your conflict of interest statement in the “Confidential to Editor” section, and submit your "Accept" recommendation.

Reviewer #1: All comments have been addressed

Reviewer #2: All comments have been addressed

2. Is the manuscript technically sound, and do the data support the conclusions?

Reviewer #1: Yes

Reviewer #2: Yes

3. Has the statistical analysis been performed appropriately and rigorously? 

Reviewer #1: Yes

Reviewer #2: Yes

4. Have the authors made all data underlying the findings in their manuscript fully available?

Reviewer #1: Yes

Reviewer #2: Yes

5. Is the manuscript presented in an intelligible fashion and written in standard English?

Reviewer #1: Yes

Reviewer #2: Yes

6. Review Comments to the Author

Reviewer #1: All changes were made, as requested. The paper is well-written and supports the growing literature that supports the routine use of mpMRI in the workup and treatment for prostate cancer.

Reviewer #2: I feel that the authors have written a good manuscript and that the paper is acceptable for publication.*

7. PLOS authors have the option to publish the peer review history of their article (what does this mean?). If published, this will include your full peer review and any attached files.

Reviewer #1: No

Reviewer #2: No

---

## [Editor Report · Acceptance letter]

1 Jul 2021

PONE-D-21-03241R1 

Risk of Adverse Pathological Features for Intermediate Risk Prostate Cancer: Clinical Implications for Definitive Radiation Therapy 

Dear Dr. Zhang:

I'm pleased to inform you that your manuscript has been deemed suitable for publication in PLOS ONE. Congratulations! Your manuscript is now with our production department. 

Kind regards, 

on behalf of

Dr. Qinghui Zhang 

Academic Editor

PLOS ONE